# Uncertainty Quantification in DL Models for Cervical Cytology

**Shubham Ojha**[1]                                         SHUBHAM.OJHA@IHUB-DATA.IIIT.AC.IN
[1] *IHub-Data, International Institute of Information Technology, Hyderabad, India*

**Aditya Narendra**[2]                                         ADINARENDRA0108@GMAIL.COM
[2] *Independent Contributor*

**Editors:** Accepted for publication at MIDL 2024

## Abstract

Deep Learning (DL) has demonstrated significant promise in digital pathological applications both histopathology and cytopathology. However, the majority of these works primarily concentrate on evaluating the general performance of the models and overlook the crucial requirement for uncertainty quantification which is necessary for real-world clinical application. In this study, we examine the change in predictive performance and the identification of mispredictions through the incorporation of uncertainty estimates for DL-based Cervical cancer classification. Specifically, we evaluate the efficacy of three methods—Monte Carlo(MC) Dropout, Ensemble Method, and Test Time Augmentation(TTA) using three metrics: variance, entropy, and sample mean uncertainty. The results demonstrate that integrating uncertainty estimates improves the model's predictive capacity in high-confidence regions, while also serving as an indicator for the model's mispredictions in low-confidence regions.

**Keywords:** Deep learning, Uncertainty, Digital Cytopathology, Cervical Cancer.

## 1. Introduction

Cervical cancer poses a substantial risk to women's health while ranking fourth in both diagnosis frequency and cancer-related mortality with over 600,000 newly reported cases and more than 340,000 fatalities globally in 2020 (Sung et al., 2021). With the advancement of DL techniques, a growing number of DL-assisted cervical cytology screening methods have emerged for tasks such as cell-level classification (Manna et al., 2021), detection (Jia et al., 2022), segmentation (Zhou et al., 2020), and whole-slide image (WSI) level diagnosis (Zhang et al., 2022). While these models excel in performance metrics, they overlook the need to incorporate methods for uncertainty estimation. Integrating these methodologies is vital for reliable clinical applications and has significant potential to improve interaction between pathologists and DL systems. For example, if a DL model provides a prediction with low confidence, it could abstain from decision-making thus enabling the pathologist to intervene in such instances. In this paper, we aim to investigate the correlation between a model's predictive capability and its uncertainty estimates for cervical cytology classification through two research questions :

1. How effective is uncertainty quantification for enhancing the model's predictive capacity in high-confidence intervals?
   We conduct a comparative analysis of a model's predictive performance with and without uncertainty quantification in high-confidence intervals.

2. Are low confidence intervals an indicator of incorrect predictions?
   Specifically, we analyse the model's predictive performance and quantify the number
   of samples associated with incorrect predictions in regions of low confidence.

## 2. Methodology

The dataset utilized in this study is derived from the cervical cell classification collection
found within the CRIC Searchable Image Database curated by the Center for Recognition
and Inspection of Cells (CRIC) (Rezende et al., 2021).In this study, we utilized a cell crop
size of 100 x 100 centred on the nucleus demarcations provided in the dataset. Using this
dataset, our objective is to develop a model for classifying cervical cells into Normal and
Abnormal categories while also estimating associated uncertainty values.

| Cell Category | Normal | Abnormal | Total |
|---|---|---|---|
| Count | 6779 | 4755 | 11534 |

Table 1: Dataset Distribution for Binary Classification

## 3. Experiments and Results[1]

This paper presents an analysis of three methods: MC Dropout, Ensemble Method, and
Test Time Augmentation(TTA), utilizing three uncertainty metrics: Entropy, Variance,
and Sample Mean Uncertainty (Pocevičiūtė et al., 2022). These methods and metrics were
assessed across three confidence intervals with thresholds = 0.3, 0.6 and 0.9 to address our
research questions.

| | MC Drop. | Ensbl. | TTA | | MC Drop. | Ensbl. | TTA |
|---|---|---|---|---|---|---|---|
| *Var.* | | | | *Var.* | | | |
| thr. 0.3 | $97.2_{\pm0.60}$ | $93.74_{\pm0.54}$ | $98.26_{\pm0.93}$ | thr. 0.3 | $26.9_{\pm0.82}$ | $38.59_{\pm3.32}$ | $18.86_{\pm5.09}$ |
| thr. 0.6 | $93.34_{\pm1.75}$ | $90.17_{\pm0.89}$ | $95.62_{\pm1.85}$ | thr. 0.6 | $34.7_{\pm2.57}$ | $41.34_{\pm7.28}$ | $27.66_{\pm5.97}$ |
| thr. 0.9 | $86.93_{\pm1.29}$ | $88.68_{\pm0.41}$ | $90.21_{\pm2.74}$ | thr. 0.9 | $42.35_{\pm7.43}$ | $25.71_{\pm23.28}$ | $36.88_{\pm6.04}$ |
| *Entr.* | | | | *Entr.* | | | |
| thr. 0.3 | $93.98_{\pm1.17}$ | $95.27_{\pm0.97}$ | $95.48_{\pm0.49}$ | thr. 0.3 | $33.71_{\pm1.77}$ | $35.5_{\pm3.33}$ | $23.01_{\pm5.28}$ |
| thr. 0.6 | $87.41_{\pm1.23}$ | $92.31_{\pm0.67}$ | $90.98_{\pm1.85}$ | thr. 0.6 | $47.66_{\pm7.28}$ | $44.1_{\pm2.35}$ | $35.76_{\pm7.01}$ |
| thr. 0.9 | $86.58_{\pm1.30}$ | $88.97_{\pm0.23}$ | $89.54_{\pm2.46}$ | thr. 0.9 | $40.00_{\pm20.00}$ | $39.99_{\pm9.94}$ | $13.33_{\pm14.14}$ |
| *S.Mean* | | | | *S.Mean* | | | |
| thr. 0.3 | $97.78_{\pm0.42}$ | $95.98_{\pm0.96}$ | $98.75_{\pm0.70}$ | thr. 0.3 | $25.01_{\pm1.04}$ | $34.16_{\pm2.96}$ | $17.07_{\pm4.67}$ |
| thr. 0.6 | $95.5_{\pm1.18}$ | $93.89_{\pm0.56}$ | $97.76_{\pm1.27}$ | thr. 0.6 | $31.72_{\pm1.28}$ | $40.09_{\pm2.17}$ | $23.94_{\pm4.88}$ |
| thr. 0.9 | $91.33_{\pm1.29}$ | $90.94_{\pm0.34}$ | $94.76_{\pm1.63}$ | thr. 0.9 | $43.24_{\pm3.51}$ | $44.26_{\pm3.32}$ | $36.14_{\pm5.68}$ |

Table 2: Comparison of AUC across High confidence intervals: (0 - 0.3), (0 - 0.6) & (0 - 0.9)

Table 3: Comparison of Mispred. rates across Low confidence intervals: (0.3 - 1), (0.6 - 1) & (0.9 - 1)

### 3.1. Implementation

We employed ResNet50 as the feature extractor for our baseline model without uncertainty
quantification and for the model with uncertainty quantification across all 3 methods. In

---

1. Link to Code : https://github.com/shubhamOjha1000/UQ-in-DL-Models-for-Cervical-Cytology

the case of MC dropout, we ran 50 stochastic passes for each input. For TTA, each input was randomly augmented 50 times before passing through the trained model. In Ensemble Method (Linmans et al., 2020), 50 MLP heads were added to the feature extractor and meta loss (Rupprecht et al., 2017) was used for better diversity during training. Further, we used min-max normalization to standardize the range of all three uncertainty metrics. Additionally, all experiments underwent 5-fold cross-validation. We then assessed performance using AUC and Misprediction rate (100 - Accuracy)% for our two research questions, respectively.

### 3.2. Results

Firstly, Table 2 indicates a 7-12 % improvement in AUC performance across all three methods in high confidence intervals compared to our baseline model, which achieved an AUC of $86.3_{\pm 2.24}$%. Additionally, Figure 1 indicates a high number of correct predictions compared to incorrect ones in low uncertainty ranges. This implies that incorporating uncertainty estimates significantly improves predictive performance in high-confidence intervals or low-uncertainty ranges. Secondly, Table 3 demonstrates that increasing thresholds for low-confidence intervals leads to an increase of 10.1-19.07 % in misprediction rates for most methods across metrics. Moreover, Figure 1 demonstrates that an increase in uncertainty range corresponds to an increased misprediction count across all metrics for MC Dropout except in Entropy. This indicates that low confidence intervals or high uncertainty ranges are indicators of incorrect predictions. However, Table 3 displays deviations from observed trends for certain methods and metrics. Moreover, Figure 1 exhibits a minimal difference between the numbers of correct and incorrect predictions, diverging from the anticipated trend of higher counts of incorrect predictions than correct ones in high uncertainty ranges. These variations question the reliability of low confidence intervals as indicators for incorrect predictions.

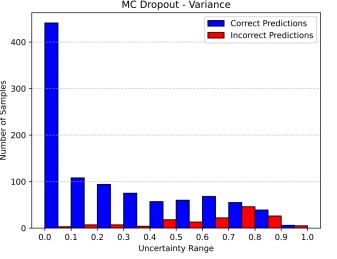 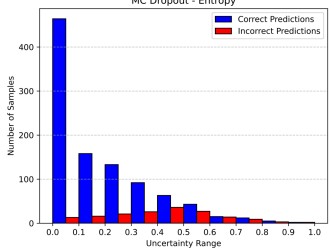 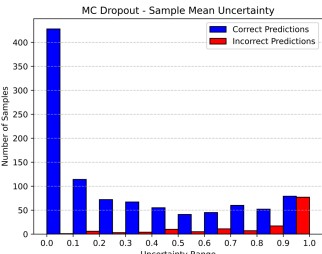

Figure 1: Distribution of Sample Predictions across Uncertainty Ranges for MC Dropout

## 4. Conclusion

In conclusion, this study demonstrates that incorporating uncertainty estimates associated with a model prediction significantly boosts the overall model performance and serves as an indicator for identifying mispredictions. Future works should focus on enhancing the robustness of the models by uncertainty integration and expanding this analysis to different datasets for the development of real-world clinical applications.

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
