# OpenReview forum: "Uncertainty Quantification in DL Models for Cervical Cytology"
_MIDL.io/2024/Short_Papers — MIDL 2024 Short Papers_

### Official Review · Reviewer_MuES · 2024-04-25

**Confidence:** 5
**Final Rating:** 5

**Review:**

This study investigates the additional value of integrating uncertainty estimates for the DL-based cervical cancer classification. Findings indicate that incorporating uncertainty estimates enhances the overall performance of the evaluated methods, enhancing the model's predictive accuracy in high-confidence regions while serving as an indicator for the model's mispredictions in low-confidence regions.

The strengths of this work are:
1) the relevance of the theme of this work, any improvement/innovation of which could have a major impact in our field
2) the relevance of the experiments and the metrics that have been chosen
3) the quality of the results

The main weakness of this article concerns the didactic aspect. The theme of uncertainty is not necessarily easy to understand for the uninitiated, and educational efforts on key points of the article (for example, the notion and influence of the choice of thresholding) should be strengthened.

---

### Decision · Program_Chairs · 2024-04-26

Accept